# The Morphological Characteristics of Authigenic Pyrite Formed in Marine Sediments

Jingyi Chang [1,2], Yuanyuan Li [1,2] and Hailong Lu [2,*]

1   College of Engineering, Peking University, Beijing 100871, China
2   Beijing International Center for Gas Hydrate, School of Earth and Space Sciences, Peking University, Beijing 100871, China
*   Correspondence: hlu@pku.edu.cn

**Abstract:** Pyrites are widely distributed in marine sediments, the morphology of which is applied as a proxy to infer the redox conditions of bottom water, and identify diagenetic stages and hydrocarbon leakage activities. In this review, the methods used for the morphological study of pyrite are summarized. The textural and size characteristics of euhedral pyrite and pyrite aggregates, as the formation and evolution mechanism of pyrite are discussed for their significance in reconstructing the geochemical environment. The morphological study of pyrite includes shape observation, size estimation, and surface feature analysis. Scanning electron microscope and optical microscope are the main methods for morphological observation; transmission electron microscope and scanning tunneling microscope are applicable to observe nanoscale morphological structures and crystal growth on the crystal surface, and X-ray computed tomography is capable of measuring pyrite size distribution at the scale of a micrometer. Under the marine sedimentary condition, the single crystal of pyrite appears in cube, octahedron, dodecahedron, and their intermediates, the size of which ranges from several nanometers to more than 100 μm. The morphology of euhedral pyrite is controlled by temperature, pH, the chemical composition of interstitial water, etc., and might have been experienced in later reformation processes. The pyrite aggregates occur as framboid, rod-like, fossil-infilling, etc., characterized by the comparatively large size of several microns to several millimeters. It is found that certain textures correspond with different formation mechanisms and geochemical environments. Particularly, under special geological conditions, for instance, the methane leakage and/or decomposition of gas hydrate, pyrite is anomaly enriched with morphological textures of massive framboid cluster, rod-like aggregates, etc., and framboid is found with a large mean diameter (>20 μm) and standard deviation (>10 μm). These typical features can be employed to ascertain the position of the paleo sulfate methane transition zone (SMTZ).

**Keywords:** authigenic pyrite; morphology; cold seep; gas hydrate; marine sediment

## 1. Introduction

Pyrite is one of the most important authigenic minerals in marine sediment [1], especially in cold seep areas, which has been the main research target in recent centuries. The growth time, as well as sedimentary rate, and geochemical condition controls pyrite formation [2], and the formation process of authigenic pyrite affects the biogeochemical cycling of iron, sulfur, oxygen, and carbon [3–6].

Previous studies revealed a sequence of morphology evolution of pyrite crystals [7] and pyrite aggregates [8], the morphological characteristics and size distribution of pyrite in normal marine sediment [9–11] and in sulfate–methane transition zone (SMTZ) [12–14], the various growth conditions during pyrite formation [15]. The morphology of pyrite provides important scientific significance for studying marine sedimentary successions, tracking microbial activities, and diagenetic processes [9,16–18].

The textures of authigenic pyrite include cubic, octahedron, pentagonal dodecahedron, framboids, sunflower (framboid inner core and external overgrowth), rod aggregates, and

organism-filling aggregates. In modern marine sediment, pyrite forms under syngenetic, diagenetic, and synsedimentary conditions, and these processes produce overgrowth and rod-like aggregates, which are the common textures found in marine sediment [13], that can be used to indicate past environmental conditions and methane-rich environments. Diagenesis makes syngenetic pyrite grow continuously, thus the morphology of previously formed pyrite may change fully or partially. Thus, pyrite in deep sediment probably has undergone complex diagenetic reformation, and pyrite in shallow sediment may record the synsedimentary environment which can be a better indicator of the sedimentary condition.

In anoxic marine sediments, organic matter and methane can react with sulfate in seawater or pore water to form hydrogen sulfide by organoclastic sulfate reduction and anaerobic oxidation of methane–sulfate reduction (AOM-SR), which then combines with iron compounds to form iron monosulfides that finally transform into pyrite ($FeS_2$) [19,20]. At seeps, methane is the predominant reactant which is oxidized by dissolved sulfate within the sediment [21]. The morphology and geochemistry of pyrite are strongly affected by methane seepage activity [13,14]. In gas hydrate-bearing sediment, the gas hydrate destabilization and methane gas expulsion led to temporally and spatially changes in seepage activity, accordingly, affecting the geochemical conditions. The AOM-derived pyrite in SMTZ has an obvious pyrite signature that shows a wide range of morphologies and mineralogical, while total sulfur (TS), total organic carbon (TOC) versus TS, sulfur isotopic characteristics are the evidence that supported to help identify SMTZ. Generally, pyrite is enriched in SMTZ, which has a wide size distribution and various textures [14,22]. Thus, this kind of pyrite is a reliable index for estimating methane flux oscillation in a methane flux fluctuate environment, the anomaly of pyrite content, texture, and size distribution is the indicator for seepage activity and gas hydrate existing.

The morphology characteristics discussed in this review concluding size, crystal shape, late alteration texture. Despite the complex diagenetic process impeding a distinct relationship between morphology and mineralization, the preliminary correlation between pyrite morphology and sedimentary environment has been proposed and discussed in this review [7].

## 2. Methods for Morphology Study

The methods usually used for studying the morphology of pyrite are optical microscope, scanning electron microscope (SEM), transmission electron microscope (TEM), scanning tunneling microscope (STM), X-ray computed tomography (X-Ray CT), etc. The resolutions of these methods are shown in Figure 1.

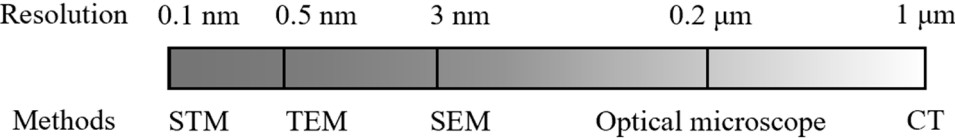

**Figure 1.** The resolution of different methods for morphological research.

Textural observation by optical microscope is the basic method for studies of pyrite texture with low cost but poor resolution (>0.2 µm). SEM is the most commonly used method for observing pyrite texture, with higher resolution (generally up to 3 nm) and relatively high cost; however, an additional pretreatment procedure is required to coat samples with Au/C/Cr. The measurement with ore microscopy may overestimate the size of pyrite in contrast to SEM results, while the latter may underestimate the size compared to the actual value with an error <10% [9]. Much higher resolution imaging can be obtained by TEM (up to 0.5 Å), which can be used to detect planar defects and thin growth layers on pyrite crystal surface [13]. However, the sample preparation process for TEM is complex and difficult (the thickness of the sample must below 200 nm), and will damage the sample [23]. STM has been adopted to investigate the micro-morphology and

structure of pyrite, which has a genetic significance and reflects various information in thermodynamics and dynamics [24].

Size distributions are commonly measured by 2D images nowadays (SEM). However, X-ray CT can provide a more convenient and in situ way substitute for SEM, especially when taking mass samples. X-ray CT was mainly used to recognize large-scale structures and study the petrophysics [25–32] in previous studies; however, X-Ray CT has been confirmed to be a useful way to calculate the size distribution of pyrite crystals [8,33–36].

## 3. Shape of Single Crystal of Pyrite

Natural euhedral pyrite crystals formed extensively in marine settings, such as normal marine sediment, marine hydrothermal environment, and microenvironment of fossils buried in marine sediment, several typical backscattered electron (BSE) images, and secondary electron (SE) images are shown in Figure 2. Except for the individual crystal, pyrite microcrystals are mostly found assembled as framboids [37].

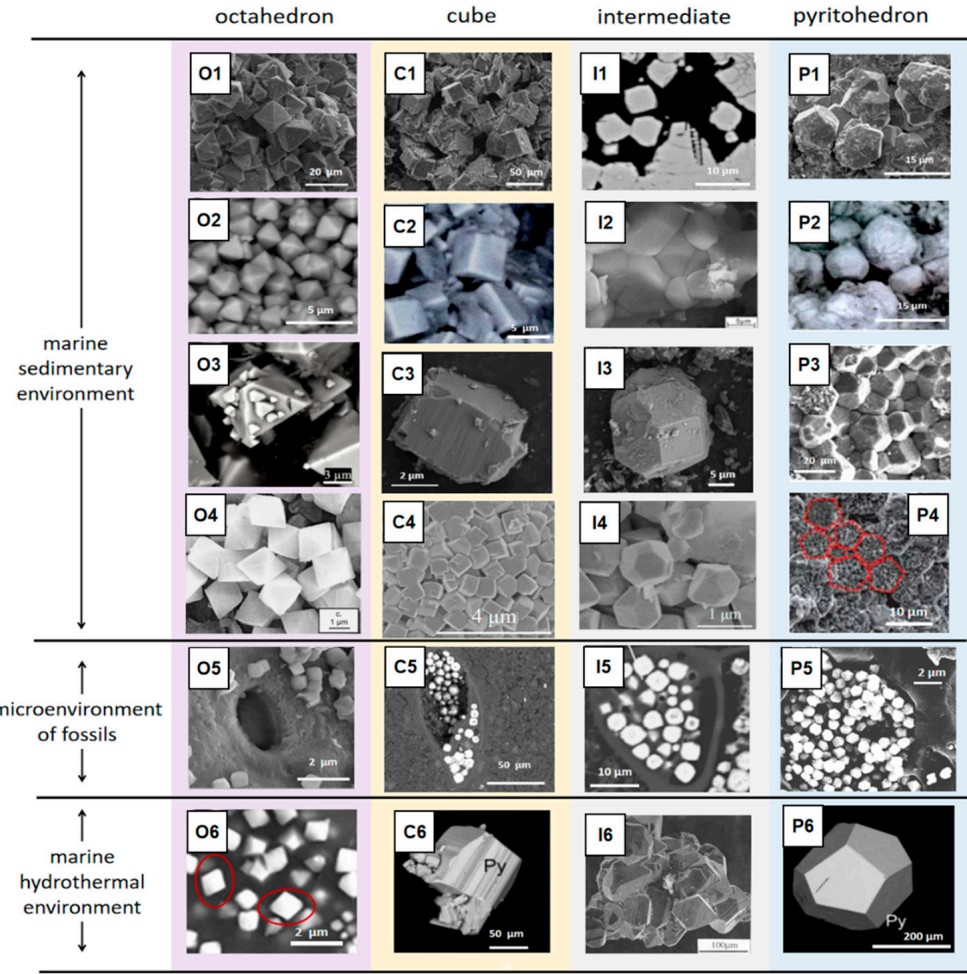

**Figure 2.** The scanning electron micrographs of euhedral pyrite in cubic form [38–42], octahedral form [8,38,43–45], pyritohedral form [12,38,43,45–47], and intermediate [13,33,44,48,49] under different marine environments, among which (**O6,C5,C6,I1,I5,P5,P6**) are BSE images, (**O1–O5, C1–C4,I2–I4,I6,P1–P4**) are SE images.

However, in the marine hydrothermal system, cube and pyritohedron are the dominant forms of authigenic pyrite, in modern marine sediment, octahedron, truncated octahedron and cubo-octahedron are the predominant forms [13,38,50,51]. Few studies have reported the cubic pyrite formation in marine sediment (<100 °C) except for seepage areas [52,53]. The summary of variations on the crystal shape of pyrite forms in marine sediment is

shown in Figure 3, there are cube, octahedron, pyritohedron, and their intermediates. During pyrite crystal growth, the combinations and morphologies of microcrystals are controlled by the geochemical growth environment, such as Fe (II) and $SO_4^{2-}$ concentration, pH, temperature, impurities, etc. According to the experimental phenomenon, Wang and Morse (1996) reported that the morphology of microcrystal transforms from cube to octahedron to spherulite texture as the increase in supersaturation of $Fe^{2+}$ and $SO_4^{2-}$ in the reaction solution [7]. With the increase in S/Fe ratio, the number of edges of microcrystalline grains also increases, pyrite crystal shape transforms from {100} or {110} + {111} to {210} + {111} [54]. The {100} becomes less stable and {111} becomes more stable with temperature increasing [55]. However, the solution pH does not seem to decisively affect the microcrystal shape transformation, and the influence of impurities on the morphology of microcrystals is still unclear [7].

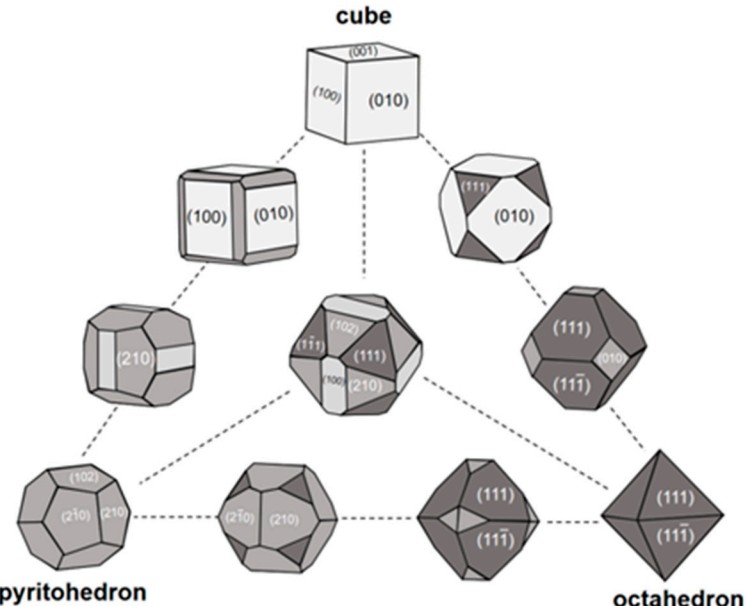

**Figure 3.** The different crystal shapes and their intermediates.

The authigenic euhedral pyrite in marine sediment mainly forms in two ways, (1) direct way, with the poor supplement of active organic matters and limited dissolved polysulfides, FeS saturation degree is hard to be reached [56,57], euhedral pyrite precipitates directly. Diagenetic or metamorphic pyrite has a larger size than sedimentary pyrite [58]. (2) Indirect way, the framboidal precursor undergoes regrowth after being buried, the euhedral pyrite is the product of diagenesis. Generally, in marine sediment, the large euhedral pyrite becomes the major texture instead of framboids, suggesting the decrease in $FeS_2$ saturation or the weak SRB activity [59] and diagenetic process [60].

Simulation experiments have also been conducted to study the morphological properties of pyrite. Alfonso used first-principles spin-polarized DFT total energy calculations to measure the stability of the crystal surface, reckoning that (001) is the preferential surface under the S-lean condition, (111) is preferred under the S-rich condition [61]. It is believed that with the increasing degree of supersaturation of iron sulfide solutions, the initially formed cube transforms into an octahedron [7]. However, Rickard [62] proposed that with initial high limiting supersaturation, (111) preferentially grows, subsequently, saturation decreases and cubic planes appear.

## 4. Texture of Pyrite Aggregates

The pyrite aggregates are usually found with the following textures, framboids, sunflower (framboid inner core and external overgrowth), rod-like aggregates [7,8,63–65].

### 4.1. Framboid

Framboid is the dominant texture of sedimentary pyrite in marine settings [62,65], it is a spherical aggregate (Figure 4A–C) composed of numerous disordered cubic or octahedral pyrite microcrystals (<2 μm) with a diameter size typically between 4 and 50 μm diameter [7,66]. Some of the framboids are densely packed, while some are scattered [67]. There is no clear spatial relationship between the octahedral pyrite microcrystals and the framboids.

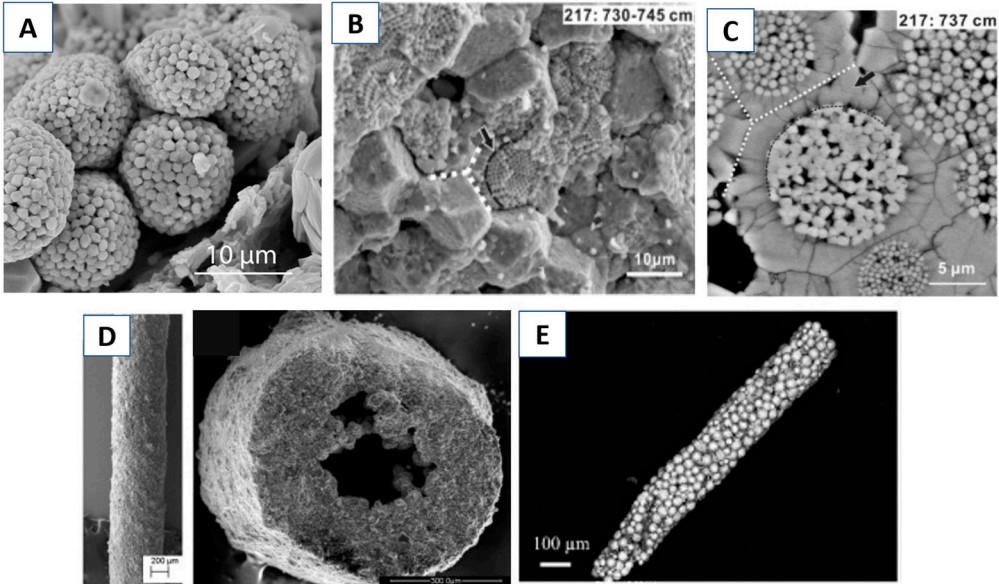

**Figure 4.** Scanning electron microscope images of different texture of pyrite. (**A**) The framboidal texture of pyrite [11]; (**B**,**C**) overgrowth around framboidal edge like a sunflower [68]; (**D**,**E**) rod-like pyrite aggregate [13,69]. Figure (**A**–**D**) are SE images and Figure (**E**) is BSE image.

There is no consensus on the formation mechanisms of spherical texture. It is primarily thought that spherical framboids inherited the structure of bacterial colonies or biogenic spherical surfaces [70,71]. However, different models have been proposed later and proved that the interaction forces such as electrostatic, gravitational, van der Waals, and magnetic forces can be the cause of aggregation [65]. Nevertheless, there is sufficient evidence showing that a high nucleation rate, and crystal growth rate ratio are necessary for framboid formation [65,72], and framboids will be preferentially formed at a high nucleation rate [72]. With a high sulfide-producing rate which makes the FeS saturation degree easier to reach, FeS will be precipitated first and transformed into framboidal pyrite later.

Three methods are used to study the formation kinetics of pyrite framboids, laboratory synthesis experiment, sedimentation rate estimation, and simulation calculation. A sedimentation rate was applied to estimate the formation time of framboids in the Peru margin for the first time [9], but the result of a long formation period is contradicted by the experimental result that pyrite framboids can be formed rapidly [73]. Rickard [74] used the diffusion–nucleation model to simulate framboid formation time and found that sedimentary pyrite framboids took around 5 days to form on average, rather larger framboids (≥80 μm in diameter) took years to form, and smaller syngenetic framboids took 3 days in average.

Framboid is commonly formed in euxinic seawater columns or shallow-marine sediments and is generally buried within a few centimeters thick sediments [75]. The particle

size and content of framboidal pyrite in modern marine sediments can reflect the sedimentary environment [75].

Framboidal pyrite formed in euxinic–anoxic bottom water environments and anoxic sedimentary environments is generally with a small average diameter and a narrow size distribution. In comparison, framboids formed in pore water of sulfidic sediment have a larger average diameter and wider size range [76]. The framboidal size distribution is indicative of a sedimentary environment, which will be discussed in detail in Section 7.

Cold seep activities, late diagenesis, low-grade metamorphism, and hydrothermal events can promote the formation of pyrite framboids [48,77,78]. Framboid clusters are ubiquitous in methane seepage or hydrate-bearing settings, which are ascribed to the additional supply of HS produced from the efficient sulfate reduction facilitated by the AOM and the sufficient iron supply from detrital iron minerals [67]. The texture of framboid clusters is controlled by sedimentary structure and pore space [14]. As the result, the co-occurrence of extremely large framboid and framboidal clusters may serve as the proxies for the strong intensity of AOM and the position of the paleo–sulfate–methane transition zone (SMTZ).

### 4.2. Sunflower

Sunflower is one of the common shapes of authigenic pyrite in marine sediment, which is formed as the overgrowth of framboids [79], the shape of which is shown in Figure 4B,C.

In late diagenesis, tabular pyrite regrows outward along the spherical rim of the inner framboidal core, then forms a subhedral (polygonal framboid) texture and finally appears in a euhedral shape [79]. As the result, the sunflower texture represents intermediate stages in the transformation of framboids into euhedral pyrites [8].

Although the framboidal pyrite that has undergone secondary growth will increase its particle size to a certain extent, the diameter of the inner core will not change, which is indicative of redox conditions [8].

### 4.3. Rod-like

The rod-like shape is the most common morphology of pyrite clusters with a size ranging from hundreds of microns to several millimeters (Figure 4D), which usually consists of abundant framboid clusters [13,80–82]. It is thought that the rod-like shape is a pseudomorph after Beggiatoa and giant filamentous bacterium [53], and another perspective is that the rod-like pyrite is formed in the gas or fluid migration channel in sediments [13,83]. Moreover, the presentation of different pyrite textures in the inner and outer parts of the rod-like aggregates [48], might indicate the different stages of pyritization or recrystallization.

To sum up, the occurrence of large quantities of rod-like pyrite may suggest the presence of methanogens, hydrocarbon migration channel, and enhanced sulfate-reducing reaction during the formation of pyrite.

## 5. Textural Evolution

The textural evolution of framboids indicates the marine sedimentary environment and geological process [18,84]. The morphology of pyrite crystal transfers from cube to cubo-octahedron and the octahedron finally forms a spherulite texture with the increase in supersaturation [7]. The morphological evolution of pyrite aggregates from framboid to euhedral has been studied by experimental and field investigations [33,75,79,85]. Generally, the following seven steps may be experienced in the formation and evolution process from iron monosulfide microcrystals to euhedral pyrites.

(1) Iron monosulfide (FeS) nucleates;
(2) FeS transforms to greigite ($Fe_3S_4$);
(3) Homogeneous greigite microcrystals aggregate to form the framboidal texture;
(4) Some greigite microcrystals grow continuously to form colloidal pyrite;
(5) The pyrites grow radially along the spherical rim of the framboid;

(6)    The framboid is conversed to subhedral and/or euhedral pyrite while the framboidal texture remains, and the voids in the subhedral stage (Figure 5F,G) disappear in the euhedral stage (Figure 5H). Finally, greigite framboids have been turned into pyrite (FeS$_2$).

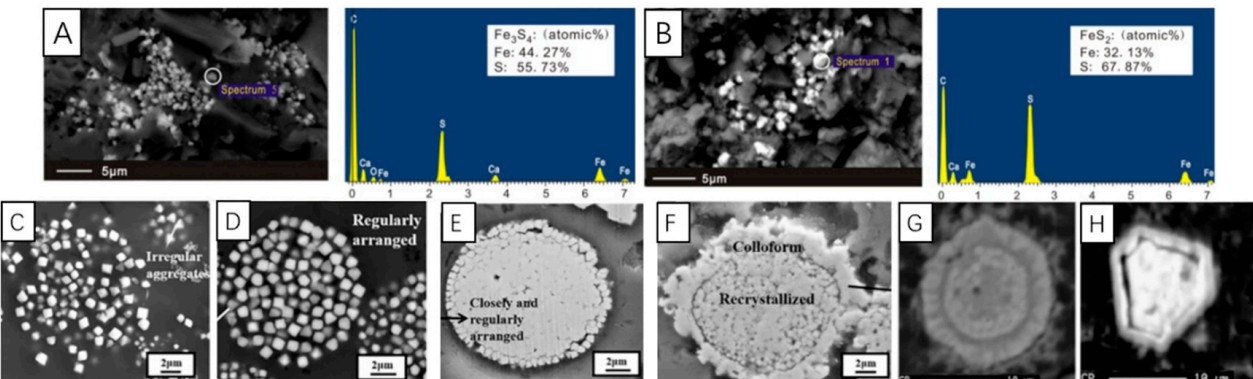

**Figure 5.** The BSE images of proposed textural evolution [33,42]. (**A**) The discrete Fe$_3$S$_4$ grain; (**B**) The discrete Fe$_2$S grain; (**C**) The disordered and irregular aggregates of pyrite microcrystals; (**D**) The regularly arranged aggregates of pyrite microcrystals; (**E**) The closely packed framboids; (**F**) The recrystallized pyrite with colloform overgrowth; (**G**) The subhedral pyrite crystal; (**H**) The euhedral pyrite crystal.

Merinero et al. [33] found the similar pyrite textural evolution in several steps in marine sediments with SEM, and a similar textural evolution was observed in modern seafloor hydrothermal system [42]. The BSE images of the proposed textural evolution are shown in Figure 5. The discrete Fe$_3$S$_4$ grain forms in marine sediment (Figure 5A) as the predecessor of Fe$_2$S (Figure 5B). The transformations from disordered and irregular aggregates of pyrite microcrystals (Figure 5C) to regularly arranged aggregates of pyrite microcrystals (Figure 5D) and closely packed framboids (Figure 5E), and from recrystallized pyrite with colloform overgrowth (Figure 4F) to subhedral (Figure 5G) and euhedral (Figure 5H) texture fit the tendency of system energy decay [42,86].

The morphology of pyrite is a significant and reliable evidence to understand the geochemical environment. The different morphologies of framboids are the reflection of different growth steps during continuous growth and textural development, and geochemical index changes simultaneously with textural evolution [33].

## 6. The Texture of Pyrite Filling in Organisms

It is quite often observed that pyrite fills in the foraminiferal chamber (Figure 6A–D) and biogenic silica (Figure 6E–H) with a texture of euhedral or framboid (shown in Figure 6A,B). The infilled texture of pyrite is observed in almost all geochemical zones in marine sediments. However, the infilled pyrite in shallow sediments usually appears in the shape of framboid, with the depth increases, degree of pyritization becomes more intensive, and infilled euhedral pyrite is formed [87]. The large euhedral crystals near the shell of foraminifera or diatom were probably formed in late diagenetic process [76,88].

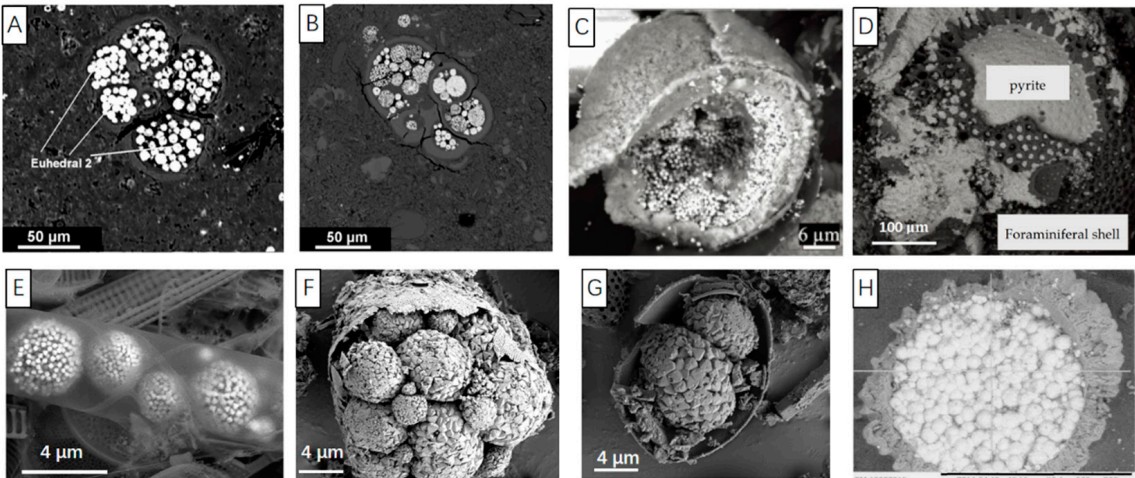

**Figure 6.** Pyrite filling in foraminifera and diatom (images from [8,40,43,89] and unpublished images from Chang). Figures (**A**,**B**) are BSE images, and Figures (**C–H**) are SE images.

The shell of foraminifera or diatom creates a comparatively confined environment restricting the material exchange between the inside and outside of the organism. This confining effect makes the geochemical environment inside the foraminiferal chamber different from that of the surrounding sediment matrix. The organic components in the diatom [90,91] and soft part of foraminifera are degraded with the involvement of the microorganism [92], and simultaneously sulfate-reducing bacteria (SRB) reduces $SO_4^{2-}$ to $H_2S$ through the metabolism [93] and accelerates the precipitation of iron sulfide. The microenvironment in the organism chamber becomes more reduced than the surrounding sediment matrix by these microbial-mediated geochemical processes. In addition, biological surfaces also act to reduce the requirement of supersaturation for Fe-sulfide formation significantly as compared with, for example, pyrite seed [94]. In the beginning stage, the pyrite forms by clinging to the inner surface of the foraminifer (Figure 6C), then grows in the outside-in pattern, and finally replaces the biological shell or grows outward through the interconnected pores (Figure 6D) [43]. It was observed that pyrite textures can be different even in adjacent fossil cells [95].

The organism-filling pyrite may reflect the reducing microenvironment in organism chamber and has certain limitations in indicating the diagenetic environment and preservation condition. However, the organic matrix is speculated as the regulatory factor during the pyritization [96] and the shape of organism-filling framboids may be related to the fossilized organic matrix-like structure [97]. The grain size of organism-filling aggregates depends on the shell size of the organisms [14]. The growth space provided by the organism is limited, thus under such an enclosed microenvironment, the growth of pyrite framboids is spatially confined and euhedral pyrite has a homogeneous size [96,98].

## 7. Pyrite Morphology in Different Geochemical Zones in Marine Sediments

Marine sediment is the most significant "sink" of iron sulfide. Framboidal pyrite can be formed in the synsedimentary process, very early diagenetic, late diagenesis, weak metamorphic process, and the hydrothermal event [99,100]. It should be noted that framboidal pyrite formed in the former two processes can be the proxy for the redox environment. Within the upper centimeters of marine sediments or euxinic water column, $H_2S$ and $Fe^{2+}$ are sufficient for the nucleation of monosulfide precursors and pyrite precipitation [66,75,101].

Based on the classic theory of early diagenetic process in sediments, organic matter/methane will be oxidized by electron receptors such as $O_2$, $Mn^{4+}$, $Fe^{3+}$, and $SO_4^{2-}$, and there is an obvious zoning sequence in the sediment depth profile. In shallow sediments, organic matter degradation coupled with sulfate reduction controls the cycling

of carbon and sulfur. In SMTZ, $SO_4^{2-}$-AOM accelerates the sulfate reduction where the product $H_2S$ promotes the pyrite precipitation, and the depth of SMTZ can shift according to the leakage intensity. Under SMTZ, metal-derived anaerobic oxidation of methane (metal-AOM) or metal-derived organic matter degradation dominates the geochemical cycle of sulfur and carbon. The pyrites occurring in different sediment zones are with obvious differences, the following mainly describes the crystal shapes and size distribution of pyrites in these three zones, organoclastic sulfate reduction, sulfate-driven anaerobic oxidation of methane, metal-derived anaerobic oxidation of methane and metal-derived organic matter degradation.

### 7.1. Organoclastic Sulfate Reduction

The pyrite in the organoclastic sulfate reduction zone is mainly formed by two mechanisms (Figure 7):

(1) Under oxic–dysoxic bottom water, the redoxcline is located under the seafloor and pyrite forms in the early diagenetic stage under the redox boundary in marine sediment. In the oxidized environment, rare framboids and pyrite crystals have been found [9,75]. In the dysoxic or anoxic environment, framboids usually with small size (mean diameter, MD < 10 μm); and narrow distribution range (standard deviation, SD < 3 μm).

(2) In the sulfidic–dysoxic bottom water column, a large amount of pyrite are formed in a relatively stable sedimentary environment with sufficient dissolved Fe, $H_2S$ and elemental sulfur. Pyrites are formed at a high growth rate and quickly sink to the anoxic sediments. After being settled to the marine sediment, pyrite cannot grow into larger due to the lack of supply of elemental sulfur. Therefore, framboidal pyrite which is formed in sulfide environment is with a small mean diameter (MD < 5 μm) and is homogeneous in grain size [10].

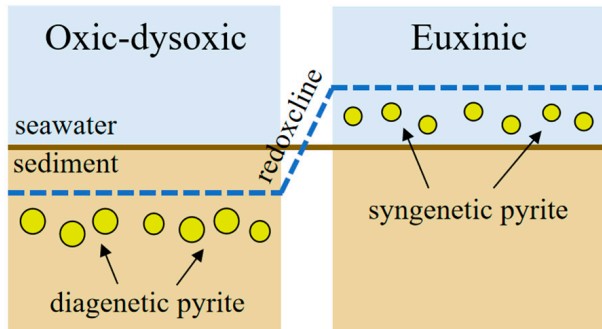

**Figure 7.** The model of pyrite formation in two different marine sedimentary environments.

The size distribution of framboidal pyrites is well investigated in modern marine sediments [22,41,102]. Most framboids range from 5 to 20 μm [9], although framboids as large as 200–250 μm have occasionally been found [103,104]. Framboidal pyrite formed in the synsedimentary, or early diagenetic stage can indicate the redox status of the depositional environment. The connection of framboid parameters with sedimentary conditions is shown in Table 1. It should be noted that the late diagenetic origin framboids (e.g., framboid aggregates as infilling of fissures and macroframboids) should be excluded from the size statistics for deduction of sedimentary condition [2].

**Table 1.** Characteristics of sedimentary framboidal pyrite used to define redox conditions (data from [6,7,24]).

| Dissolved Oxygen (mL/L) | Conditions | Framboid Parameters | | | |
|---|---|---|---|---|---|
| | | Mean Diameter (µm) | | | Texture |
| | | [105] | [9] | [11] | |
| 0 | Euxinic | 3–5 | 3–6.7 | 2.9–10.9 | Framboid dominant |
| 0–0.2 | Anoxic | 4–6 | | | Framboid dominant |
| 0.2–2 | Lower dysoxic | 6–10 | 3.3–11.8 | 3–20.9 | Large framboid and some crystalline pyrite |
| | Upper dysoxic | <5 | | | Pyrite crystal dominant |
| 2.0–8.0 | Oxic | | | | No framboid, rare pyrite crystals. |

*7.2. Sulfate-Driven Anaerobic Oxidation of Methane*

Framboid is the most common texture formed in methane seepage areas. About 50% of sulfate is fixed in authigenic pyrite through the AOM-SR process, which is 10 times higher than sulfate fixed by anaerobic oxidation of organic matter [93,106,107]. The following summarized pyrite characteristics in SMTZ.

(1) A great number of authigenic pyrites are formed in SMTZ, especially framboidal pyrites, because of the abundant organic matter and $H_2S$ supply. The content of pyrite is obviously higher in SMTZ of hydrate occurring area than that in sediment without gas hydrate.

(2) The size of the framboid is with a tendency to be larger (mean sizes >20 µm) and more variable in size (standard deviations >3.0 µm).

(3) Most framboids form clusters, and sometimes overgrowth can be observed. Some of the framboids appeared in the shape of vertically oriented rods, which might represent the migration pathways of the strong flux of methane in sediments.

In summary, large framboidal size is generally associated with the anomaly high pyrite content. As the result, larger framboids in sediment might suggest an enhanced AOM activity [14].

It is found that pyrite formed by dissimilatory sulfate reduction is usually buried in the shallow sediment, which is easy to be re-oxidized by bioturbation and turbidity activities [108,109]. On the contrary, the SMTZ at deeper depth is in a condition of less oxidizing as the oxygen from oxidized bottom seawater is difficult to reach [106]. Authigenic pyrite accumulated in SMTZ of gas-hydrate bearing sediment might have been stable for at least ~4400 years before the present [106].

However, in recent marine sediment, framboids have been found both under sulfidic (e.g., modern Black Sea [20,110]) and oxic (e.g., the South China Sea [48,80]) environments. It is because, in cold seep or gas hydrate-bearing sediment, the methane flux or gas hydrate destabilization may induce an active seepage event. With sufficient reactive Fe, CH4, and strong microbial-mediated AOM-SR, pyrites grow continuously and there is a size enhancement for framboid. As shown in Figure 8, synthetic pyrite framboid formed in the euxinic condition is with a relatively narrow size distribution (MD < 20 µm; SD < 5 µm), diagenetic framboid formed under the oxic–dysoxic condition without the contribution of AOM is with a larger and more variable size (MD < 20 µm; SD < 5 µm) [111], framboidal pyrite formed in the SMTZ generally has a large mean diameter (15–80 µm) and a wide size range (3–35 µm) [14,22,33]. The former two (synthetic framboid and diagenetic framboid without AOM participation) usually formed in single framboids and framboids formed in the SMTZ probably formed as clusters [14]. There is no apparent distribution pattern of framboidal size buried in SMTZ. However, some framboidal pyrite reported in normal marine sediment also has an SD larger than 5 µm [111], half of the framboids in gas hydrate-bearing layers measured by Lin Q et al. and Miao et al. [14,22] have an SD less than 5 µm.

Hence, the increment of the diameter of framboids may be a novel proxy than SD for distinguishing AOM enhancement.

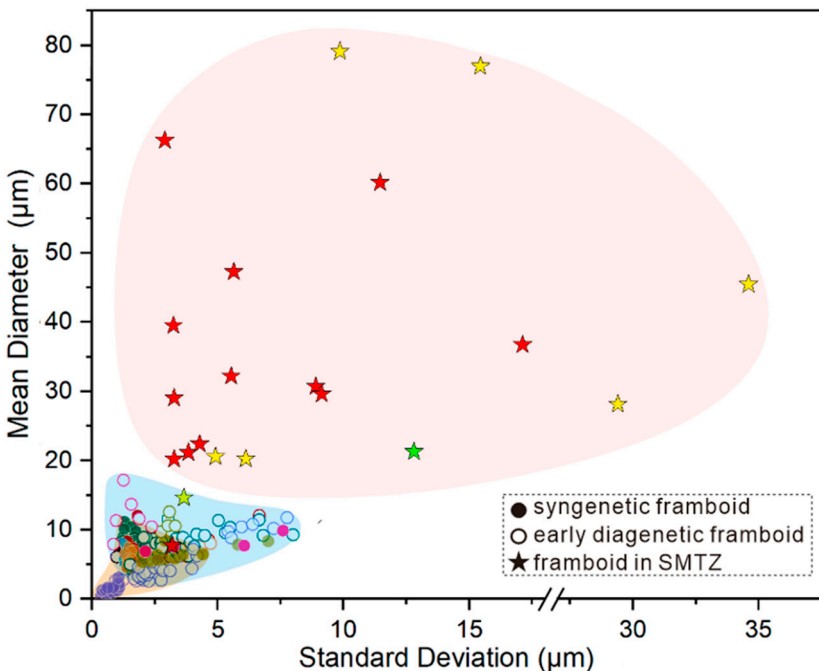

**Figure 8.** The mean diameter and standard deviation of framboidal pyrites form in different marine environment. The data of syngenetic framboid is from [9,101,104,105], early diagenetic framboid is from [14,22,111] and framboid in SMTZ is from [14,22,33]. The orange shaded area is syngenetic framboid, the blue shaded area is early diagenetic framboid and the pink shaded area is framboid in SMTZ.

The size enhancement effect is not only reflected by large authigenic framboids but also is simultaneously illustrated by overgrowth layers on previously formed pyrite associated with organoclastic sulfate reduction in SMTZ [14]. As the result, it is noted that the mean diameter and standard deviation of framboidal pyrite formed by AOM in cold seep or gas hydrate-bearing sediment are usually larger than the framboids produced by organoclastic sulfate reduction. Consequently, the anomalous size of framboidal pyrite can serve as an index for methane leakage activity.

### 7.3. Metal Driven Anaerobic Oxidation of Methane

The sulfate is depleted below SMTZ, where metal-driven anaerobic oxidation of methane (metal-AOM) and metal-driven organic matter degradation (metal-OMD) will be the dominant pathways of pyritization [112–115]. Because of the limited burial depth and low activity of residual TOC, metal-derived organic matter degradation could be largely restricted in deep marine sediment [113,116]. The iron-driven AOM occurs at the one-tenth rate of sulfate-driven AOM although it is more energetically favorable [115], which could be attributed to the easier reaction accessibility of sulfate than solid metal oxides. Therefore, the formation rate of pyrite in the metal-AOM zone can be much lower than that in SMTZ accordingly, framboidal pyrite in the metal-AOM zone is supposed to have a relatively small size as compared with euhedral pyrite. However, there are still many questions to be addressed about the material supply, and microbial participation mechanism of metal-AOM, and the studies on pyrite morphology within the metal-AOM zone are few, so further investigations are needed.

In summary, as shown in Figure 9, there are no framboids and rare crystalline pyrites buried in oxic marine sediments. Under the dysoxic condition, both diagenetic euhedral pyrites and framboids occur in shallow sediments, and pyrite framboids might have a broad

size range. It is found that the synthetic pyrite framboids formed in a euxinic environment are generally small in size and have less size variation. In the zone of organoclastic sulfate reduction, pyrite framboids scatter individually and some of them have corrosion on the crystal face. Authigenic pyrite in SMZT generally occurs in framboids and with overgrowth to the larger diameter (15–80 μm), such as framboid clusters and rod-like aggregates even with a size of several millimeters. The characteristics of organism-filling pyrite are associated with the microenvironment condition in a foraminiferal, diatom, or radiolarian shell, which cannot be a proxy for a sedimentary environment. The rate of pyrite formation in the metal-AOM zone is much lower than that formed in SMTZ, consequently, the pyrite formed in the metal-AOM zone might have a relatively small size.

**Figure 9.** Morphological patterns of authigenic pyrite under different redox condition in marine sediment.

## 8. Alteration in Later Diagenetic Processes

In the process of late diagenesis, it is generally thought that the regrowth of pyrite framboids is limited, so the morphology and size of framboids remain stable [9]. Once formed, only the infilling of organism shell, overgrowth or recrystallization can change pyrite framboid over the geologic time period, so the size distribution of framboids is a reliable and robust proxy for the understanding of the redox condition of ancient bottom water and deciphering the sedimentary environment evolution at exceptionally high precision [2,14,107–109]. As the result, pyrite framboid can be used to interpret the evolution of the sedimentary environment [17], and this method is even applicable to a weathered sample that retains framboid pseudomorph [101]. Oxidized framboid which retains its original shape and size still has environmental significance.

### 8.1. Oxidation

8.1.1. Abiotic Oxidation

Framboid is not the most stable form of pyrite. Under certain conditions, framboids can be transformed into a euhedral shape, such as a truncated octahedron and/or pyritohedron.

Oxidation plays a vital role in affecting the Fe/S cycle and also changing the morphology of pyrite. Numerical simulations and laboratory experiments are applied to study the oxidizing behavior and oxidation mechanism [117–119], and observations on natural pyrite crystals have also been conducted. It is found that redox condition, lattice

structure, impurities, and imperfections of pyrite crystal are the factors controlling the oxidation degree.

The oxidation of pyrite is an electrochemical process initiated by the adsorption of $O_2$ and $H_2O$ on the surface Fe sites [120]. It is suggested that the surface atomic configuration controls the water dissociation and electron transfer reactions, so the oxidation rate of nano/microcrystals of pyrite with different morphologies is unequal. The initial oxidation rates of pyrite (210) and (111) are 10–100 times greater than (100). In high humidity air, the initial oxidation rate of pyrite (111) is greater than (210) [119].

As shown in Figure 10A, in laboratory conditions, no matter whether in a reducing or oxidizing environment, the light grey oxidized zone of {111} is larger than {100}, indicating that {111} is easier to be oxidized. Figure 10B illustrates the difference in oxidation degree on the different crystal faces of a truncated octahedron in the natural environment. On this crystal, it can be seen that (100) is smooth and (111) has pits on it, representing better stability of (100) than that of (111). As the result, pits on pyrite planes might indicate the experience of oxidation. Based on experiment results [117], oxidation of (100) can happen only when it was experienced an even stronger oxidizing process.

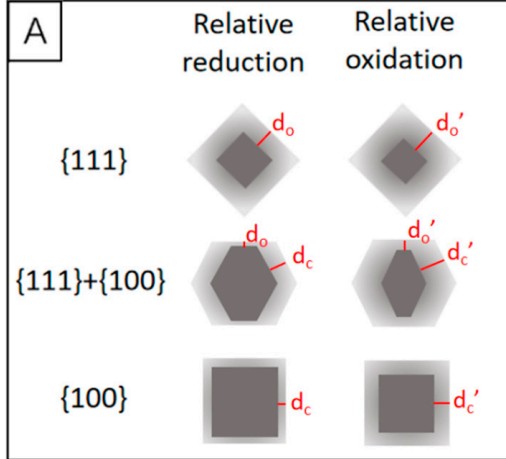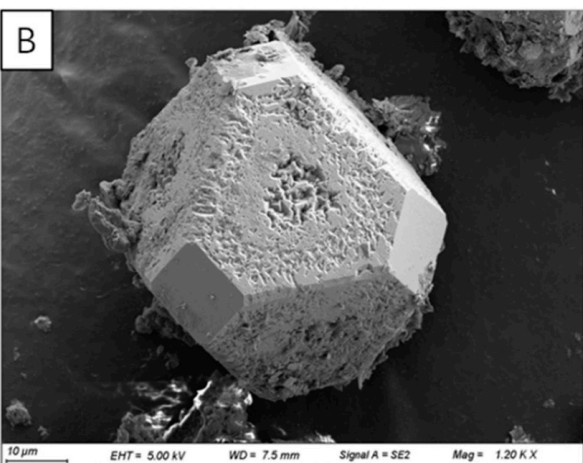

**Figure 10.** (**A**) The models of oxidized pyrite with different morphology under two redox conditions modified from [117], the light gray zone is the oxidized shell and the dark gray zone is the unoxidized pyrite core, $d_o$ and $d_c$ are the thickness of the oxidation shell along the [111] and [001] direction. (**B**) The SE image of differential oxidation on different crystal faces of a truncated octahedral pyrite.

Generally, the impurity level and the structural imperfections are the principal factors accelerating the oxidation process [121]. The degree of accumulated impurities in a single crystal is in a larger range than that in aggregates, and the low limit of impurities level in a single crystal is approximately equal to the top limit of impurities level in pyrite aggregates [121].

### 8.1.2. Biological Oxidation

Microbe-mediated oxidation of pyrite is realized by adhering to the pyrite surface and subsequent enzymatic reactions [122]. The biofilms growing on pyrite with localized enrichments of nitrogen, the biologically mediated microstructures such as pits and channels on pyrite surfaces, and layers of Fe-oxides on pyrite surfaces are supportive evidence of biological dissolution.

A two-year-long laboratory experiment shows that mainly three patterns of biological corrosion can be developed on pyrite surface, isolated corrosion pits, pearl-string-like chains, and channel-like corrosion structures [123], which are consistent with the etched patterns of pyrite in sediments. The scattered corrosion pits in rounded/elongated shapes distribute near bacteria, with a size slightly larger than bacteria. The chains of corrosion pits can be formed in various curved shapes, which might be developed faster in the crystal

with dislocation crystal than in the undisturbed area of a regular crystal. By chemical corrosion, a chain-like structure is easily transformed into a channel-like structure, and faint pits can be seen under the sunken channel. The SE images of three kinds of bacterial corrosion patterns on pyrite surfaces are shown in Figure 11.

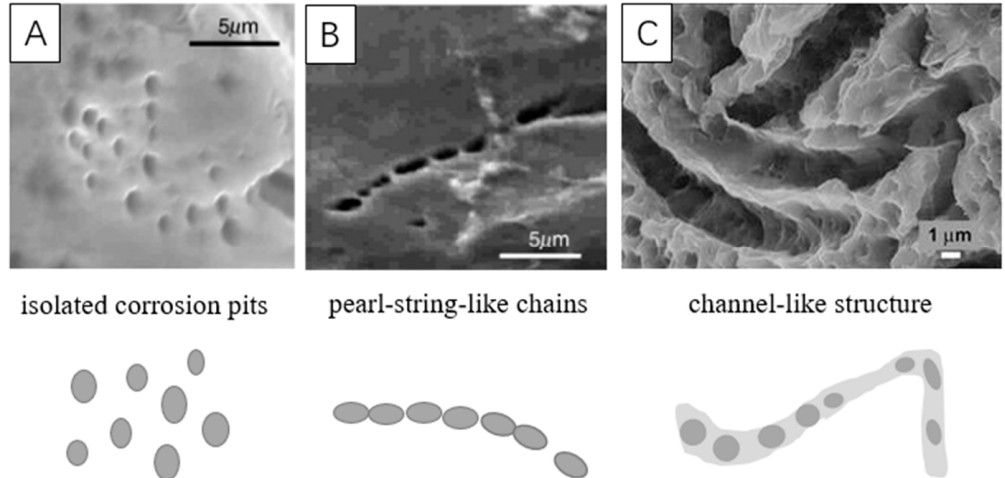

**Figure 11.** Bacterial corrosion patterns on pyrite surfaces. (**A**) isolated corrosion pits [124]; (**B**) pearl-string-like chains [124]; (**C**) channel-like corrosion structure [125].

Huang et al. (2019) did a lot of analyses and comparisons between framboid and oxidized framboid through weathering, which confirms that the oxidation process would not shrink the size of framboid and accordingly has a negligible impact on the interpretation of the redox state of the ancient ocean [126]. Nevertheless, the geochemical information recorded in pyrite could be the result of multiple origins during strong weathering. Therefore, analyzing the geochemical data of this type of sample requires careful consideration.

*8.2. Recrystallization and Overgrowth*

It is believed that there is a strong genetic link between framboid and euhedral pyrite. The euhedral pyrite grain may be the product of crystal lattice oriented overgrowth on existing textural types [99]. The overgrowth layer requires a longer growth time than framboidal pyrite, indicating a continuous $H_2S$ supply by diagenetic process. Sawlowicz proposed three hypothetical evolutional pathways [79] (Figure 12):

(1)  Tabular pyrite regrows outward along the spherical rim of inner framboid, gradually forms a subhedral (polygonal framboid) texture and finally a euhedral texture [79]. Although the framboidal pyrite that has undergone secondary growth will increase its particle size to a certain extent, the diameter of the inner core will not change, which is indicative of redox conditions.

(2)  Microcrystals in framboids continuously grow and the internal pores may be filled by newly formed pyrite and colloidal pyrite, the framboid transforms into solid spherical pyrite, which may further evolve into euhedral pyrite.

(3)  When the internal material of the framboid is plastic enough to move, the regular euhedral faces directly form via polygonal framboid.

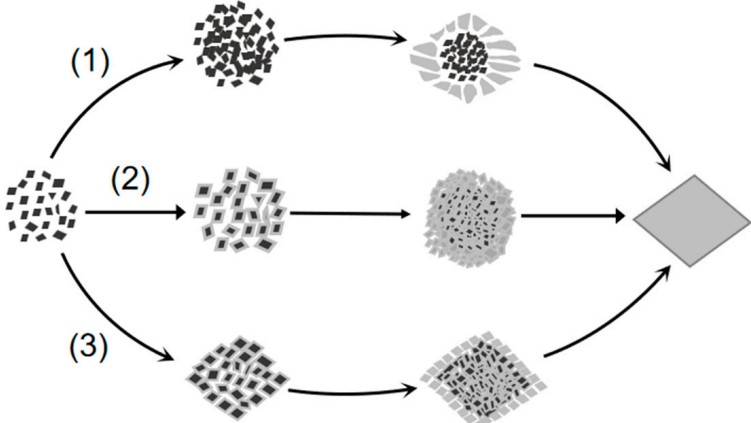

**Figure 12.** Hypothetical pathways for the formation of euhedral pyrite via framboids (modified after [79]). The pathways of (1) (2) (3) correspond to the three hypothetical evolutional pathways stated above.

*8.3. Redeposition*

The redeposition mechanism was proposed on the basis of the contradiction between the anoxic environment indicated by enriched framboids and the highly oxygenated conditions by abundant gastropod and ostracod shells [127]. Additionally, special sedimentary structures related to such a process can help to determine the occurrence of redeposition. The pyrite framboid may not be deposited directly to the seafloor after formation but suspends by attaching to an organic matter [9], moves with upwelling and advective circulation to the upper layer of the sea, and finally is deposited and buried in other marine sediments.

Therefore, it should be confirmed first that the pyrite forms in situ before using it to study the paleo geochemical environment and diagenetic process. Moreover, the morphological study of pyrite needs to consider various geochemical indexes to avoid the misunderstanding of the marine environment made by the deviation of a single indicator.

**9. Summary**

The morphological characteristics of pyrite, including the shape of crystal or crystal aggregate, texture, and size distribution, can be applied as proxies for studying the geochemical environment and diagenetic process of marine sediments.

Octahedron is identified as the dominant crystal shape of authigenic pyrite in marine sediments, while the cubic form is relatively less observed. Framboid, framboidal cluster, and rod-like aggregate are typically found in the sulfate–methane transition zone (SMTZ), generally being associated with intensive anaerobic oxidation of methane (AOM). In the diagenetic process, pyrite microcrystals and framboids might have been transformed into larger euhedral form.

Pyrite morphologies in different geochemical zones in marine sediments are different. In the zone of organoclastic sulfate reduction, the morphological characteristics of pyrite are controlled by the redox condition of bottom seawater and generally with small size (<20 μm); in sulfate–methane transition zone (SMTZ) authigenic pyrite generally occurs in the form of framboid cluster and/or rod-like aggregate and with a diameter of 15–80 μm; in metal-AOM zone pyrite is rarely formed.

The statistical results of the size of framboidal pyrite have been widely employed for studying the paleo marine environment. The mean diameter and standard deviation of framboidal pyrite are generally used to determine the redox condition of bottom water. The values of mean diameter (>20 μm) and standard deviation (>3 μm) of framboidal pyrite can be used to identify methane leakage and/or decomposition of gas hydrate, and mean diameter might be a more sensitive parameter than the standard deviation.

It should be noted that the morphological characteristics of pyrite in marine sediments might have been reformed in the process of post-deposition, by the reactions such as oxidation, recrystallization, etc. As a result, it is better to comprehensively consider other mineralogical, geochemical, and sedimentological data when applying morphological information about pyrite to reconstruct the marine environment.

**Author Contributions:** Conceptualization, H.L.; writing—original draft preparation, J.C.; writing—review and editing, Y.L. All authors have read and agreed to the published version of the manuscript.

**Funding:** DD20221703 from the China Geological Survey; The Guangdong Major project of Basic and Applied Basic Research (No.2020B0301030003).

**Conflicts of Interest:** The authors declare no conflict of interest.

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
