# Peer review of "The Morphological Characteristics of Authigenic Pyrite Formed in Marine Sediments"

_jmse, doi:10.3390/jmse10101533_

Round 1

Reviewer 1 Report

The manuscript Chang et al. presents a detailed review of how morphology of pyrite is related to marine sediments environment. They explain the methods to study the morphologies, the shape of the morphologies, texture and how the texture is related to environment, diagenesis, recrystallization, alterations and corrosion. I have found the paper interesting and easy to read. I retain that this is a nice piece of science that need to be published and will probably receive several citations. There are some typos and few sentences that needs to be rewritten. But the main issue is that the references are a mess. Although they cite the most important papers on the topic and the reference list is fine with me, they misplaced their position and they do not recall the right paper in the right place. It is really confused on that. 

The paper is very interesting and deserve publication but only when the reference will be organized and they apply all corrections.

Some examples of messy references:

§  REFERENCE 42: Coleman, M.L.a.R., R. Source of carbonate and origin of zonation in pyritiferous carbonate concretions; evaluation of a dynamic model. American Journal of Science 1995, 295, 282-308

Check reference [42] in lines 45 to 49 of the manuscript. The article in ref. 42 deals with carbonate and pyrite concretions in Toarcian (Jurassic) black shales and not with modern marine sediments.

§  REFERENCE 47 e 48:

47. Boetius, A.; Ferdelman, T.; Lochte, K.J.D.-S.R.P.I. Bacterial activity in sediments of the deep Arabian Sea in relation to vertical flux. 2000, 47, 2835-2875.

48. Hinrichs K. U., S.R.R., Müller P. J. and Rullktter J. A biomarker perspective on paleoproductivity variations in two late quaternary sediment sections from the southeast atlantic ocean. Organic Geochemistry 1999, 30, 341-366.

Check references [47,48] in lines 59 and 60 of the manuscript. The article in refs. 47 and 48 are not about methane seeps. 

§  From lines 60 to 69 there is no reference.

9. Zhang, J. The coupling of carbon and sulfur in sediments in the early diagenesis of methane hydrate potential area of northern South China Sea. Xiamen University, 2014.

What is it? Master or PhD Thesis or what else?

11. Chen Huichang, L.Y., Lu Hailong, Liang Jinqiang, Lu Jingan, Fang Yunxing. Study on authigenic pyrite in sediments of gas hydrate geo-system in the Shenhu area, South China Sea. Haiyang Xuebao 2018, 40, 116-133.

Wrong order

36. Wang, Q.; Morse, J.W.J.M.C. Pyrite formation under conditions approximating those in anoxic sediments I. Pathway and morphology. Marine Chemistry 1996, 52, 99-121. 

Wrong names

70. E., R.C. A steady-state model for sulphur isotope fractionation in bacterial reduction processes. Geochimica et Cosmochimica Acta 1973, 37, 1141-1162, doi:10.1016/0016-7037(73)90052-5.

Wrong names

79. L., W.R.T.a.B.H. Pyrite formation by reactions of iron monosulfides with dissolve dinorganic and organic sulfur species. Geochim. et Cosmochim Acta 1996, 60, 4167-4179.

Wrong names

99. HUDSON, J.D. Pyrite in ammonite-bearing shales from the Jurassic of England and Germany. Sedimentology 1982, 29, 639-667, doi:https://doi.org/10.1111/j.1365-3091.1982.tb00072.x 

Name with capitals

108. Palomares, R.M., Hernández, R. L., González-Sanz, F. J., Losada, L. S., and Martínez-Frías, J. A Mathematical Algorithm to Simulate the Growth and Transformation of Framboidal Pyrite: Characterization of the Biogenic Influence in Their Size Distributions. In Mathematics of Planet Earth; Springer: Berlin, Heidelberg., 2014. 

Age at the end.

121. J., B.E.; H., H.C.; J., O.V. Manganese- and Iron-Dependent Marine Methane Oxidation. Science 2009, 325, 184-187.

Wrong names

Many references are missing the DOI.

The minor issues are:

Line 154: However, different models have been conducted later indicate 154 that interaction forces such as electrostatic, gravitational, van der Waals, magnetic forces 155 can be an explanation for cause of aggregation [79].

Something is missing in this sentence. Check “indicate”.

Line 256: check the verb “peculated”.

Line 263: Chenge maine to “marine”

Line 271: Check the “:” at the end of the sentence.

Line 273: define “C-S cycle”.

Line 276: check “dominate”.

Line 358: I think it is correct in the following way: “ in metal-AOM zone is supposed to have a relatively small size”.

Line 363: I think it is correct in the following way: “ both diagenetic euhedral pyrites and framboids form in shallow sediment”

Line 373: I think it is correct in the following way: “ the pyrite is supposed to have a relatively small size”.

Line 393: I think it is correct in the following way: “There are several computer simulations and laboratory experiments that are applied to study the oxidative behavior and oxidation mechanism [26,123,124].”

Line 403: I think it is correct in the following way: “As shown in Figure 10A, whether in a reduction or oxidation environment, the red zone of 111} is larger than 100}, and 111} is the more redox-sensitive pyrite shape.

Line 407: “The former surface is smooth”…

Line 409: “intense or the late stage of oxidation process,”

Line 429: check “crystallographically”

Line 430: “It is easy for corrosion chains further develop to channel-like structures by chemical corrosion, faint pits can be seen under the sunken channel.” Divide sentence.

Line 453: “ (2) Microcrystals in framboid continuously growth,” remove “(1)”.

Line 457: what is “uty.”?

Author Response

  • REFERENCE 42: Coleman, M.L.a.R., R. Source of carbonate and origin of zonation in pyritiferous carbonate concretions; evaluation of a dynamic model. American Journal of Science 1995, 295, 282-308

Check reference [42] in lines 45 to 49 of the manuscript. The article in ref. 42 deals with carbonate and pyrite concretions in Toarcian (Jurassic) black shales and not with modern marine sediments.

  • REFERENCE47 e 48:
  1. Boetius, A.; Ferdelman, T.; Lochte, K.J.D.-S.R.P.I. Bacterial activity in sediments of the deep Arabian Sea in relation to vertical flux. 2000, 47, 2835-2875.
  2. Hinrichs K. U., S.R.R., Müller P. J. and Rullktter J. A biomarker perspective on paleoproductivity variations in two late quaternary sediment sections from the southeast atlantic ocean. Organic Geochemistry 1999, 30, 341-366.

Check references [47,48] in lines 59 and 60 of the manuscript. The article in refs. 47 and 48 are not about methane seeps. 

Response: Thanks for pointing out this, the references have been checked through and corrected accordingly.

  • From lines 60 to 69 there is no reference.
  1. Zhang, J. The coupling of carbon and sulfur in sediments in the early diagenesis of methane hydrate potential area of northern South China Sea. Xiamen University, 2014.

What is it? Master or PhD Thesis or what else?

Response: It is a mater thesis of Jie Zhang, and the citation format has been corrected.

  1. Chen Huichang, L.Y., Lu Hailong, Liang Jinqiang, Lu Jingan, Fang Yunxing. Study on authigenic pyrite in sediments of gas hydrate geo-system in the Shenhu area, South China Sea. Haiyang Xuebao 2018, 40, 116-133.

Wrong order

  1. Wang, Q.; Morse, J.W.J.M.C. Pyrite formation under conditions approximating those in anoxic sediments I. Pathway and morphology. Marine Chemistry 1996, 52, 99-121. 

Wrong names

  1. E., R.C. A steady-state model for sulphur isotope fractionation in bacterial reduction processes. Geochimica et Cosmochimica Acta 1973, 37, 1141-1162, doi:10.1016/0016-7037(73)90052-5.

Wrong names

  1. L., W.R.T.a.B.H. Pyrite formation by reactions of iron monosulfides with dissolve dinorganic and organic sulfur species. Geochim. et Cosmochim Acta 1996, 60, 4167-4179.

Wrong names

  1. HUDSON, J.D. Pyrite in ammonite-bearing shales from the Jurassic of England and Germany. Sedimentology 1982, 29, 639-667, doi:https://doi.org/10.1111/j.1365-3091.1982.tb00072.x 

Name with capitals

  1. Palomares, R.M., Hernández, R. L., González-Sanz, F. J., Losada, L. S., and Martínez-Frías, J. A Mathematical Algorithm to Simulate the Growth and Transformation of Framboidal Pyrite: Characterization of the Biogenic Influence in Their Size Distributions. In Mathematics of Planet Earth; Springer: Berlin, Heidelberg., 2014. 

Age at the end.

  1. J., B.E.; H., H.C.; J., O.V. Manganese- and Iron-Dependent Marine Methane Oxidation. Science 2009, 325, 184-187.

Wrong names

Many references are missing the DOI.

Response: The mistakes above have been corrected.

The minor issues are:

Line 154: However, different models have been conducted later indicate 154 that interaction forces such as electrostatic, gravitational, van der Waals, magnetic forces 155 can be an explanation for cause of aggregation [79].

Something is missing in this sentence. Check “indicate”.

 Response: The wordings mentioned by the reviewer have been checked and changed.

Line 256: check the verb “peculated”.

Response:It has been changed as to the suggestion of the reviewer.

Line 263: Chenge maine to “marine”

Line 271: Check the “:” at the end of the sentence.

Response: It has been corrected as to the comment of the reviewer.

Line 273: define “C-S cycle”.

Response: Thanks for the comment, C-S cycle has been changed to cycling of carbon and sullfur

Line 276: check “dominate”.

Line 358: I think it is correct in the following way: “ in metal-AOM zone is supposed to have a relatively small size”.

Line 363: I think it is correct in the following way: “ both diagenetic euhedral pyrites and framboids form in shallow sediment”

Line 373: I think it is correct in the following way: “ the pyrite is supposed to have a relatively small size”.

Line 393: I think it is correct in the following way: “There are several computer simulations and laboratory experiments that are applied to study the oxidative behavior and oxidation mechanism [26,123,124].”

Line 403: I think it is correct in the following way: “As shown in Figure 10A, whether in a reduction or oxidation environment, the red zone of 111} is larger than 100}, and 111} is the more redox-sensitive pyrite shape.

Line 407: “The former surface is smooth”…

Line 409: “intense or the late stage of oxidation process,”

Response:Thanks for the comments, and they have been revised according to the above suggestions.

Line 429: check “crystallographically”

Response: Indeed it is easy to be confused, so it has been changed to in the undisturbed area of a regular crystal.

Line 430: “It is easy for corrosion chains further develop to channel-like structures by chemical corrosion, faint pits can be seen under the sunken channel.” Divide sentence.

Response:The sentence has been changed as to the suggestion.

Line 453: “ (2) Microcrystals in framboid continuously growth,” remove “(1)”.

Line 457: what is “uty.”?

Response: Sorry, it is a typo, so uty has been deleted.

Reviewer 2 Report

The manuscript of Jingyi Chang, Yuanyuan Li  and Hailong Lu. “ The morphological characteristics of authigenic pyrite formed in marine sediments” provide an interesting review on pyrite varietis, their  morphology, sizes, rate of growth and oxidation in different environments. It is shown pyrite evolution from framboidal to euhedral varieties. Microbial traces images are useful for geobiology development. It is very important that the authors elaborated a new criteria for recognition of paleo sulfate-methane transition zone: the methane leakage and decomposition of gas hydrate, py rite is anomaly enriched with morphological textures of massive framboid cluster, rod-like aggregates, etc., and framboids are found with large mean diameter (> 20 μm) and standard deviation (> 10 μm). They confirmed previous result of correlation between the particle size of pyrite framboids and their aggregates with this zone distribution. The general results displayed in the paper have  important theoretical and practical significance of pyrite morphology to be used. I think, this is a well written, illustrated and insightful study that provides some hypothethis about origin of the deposit. However, I have a technical remark: all abreviation like “OSR, TS, TOC, SMI” and others could be deciphered in one group in Introduction for pleasure of a wide range of the readers. The text has some overprint mistakes.The manuscript can be accepted with minor revision.

Author Response

Response: We do appreciate the comments of the reviewer. The TOC, TS have been defined in the section of Introduction for better understanding. And SMI has been substituted by SMTZ for being consistent through this manuscript.

Reviewer 3 Report

The paper by Jingyi Chang and coauthors presents a review on morphological characteristic of authigenic pyrite, which forms in marine sediments. In general, the paper is rather well structured, contains a lot of necessary references, emphasized the most distinctive morphological features of marine authigenic pyrite and deserves to be published in Journal of Marine Science and Engineering.

It will be nice if the authors also consider the geochemical peculiarities of framboidal pyrite (e.g., trace contents or S isotopic composition) from various environments, since it definitely can be a proxy of the formation condition. In addition, the authors also must carefully re-read their paper to avoid numerous inconsistencies and typing errors (some of them are indicated in the pdf file of the paper).

Author Response

Response: We do appreciate the suggestions of the reviewer. In this manuscript, the discussion is focused on the morphology of pyrite and it is significance in studying the geochemical conditions. However, we are going to write a review on the changes of geochemical parameters in the formation of pyrite.

Reviewer 4 Report

General comments:

This review collects a huge amount of published papers from the field of framboidal pyrite research. however, the aims and the novelty what this collection brings, are not clear. The conclusions, which can be drawn based on the review of these papers, are not well emphasised or even not clearly drawn.  Please, find it out, how could You better emphasise Your added value, maybe a more simple structure for the whole manuscript would also help. It should contain more summarising figures and tables for easier understanding and usage. In its present form, this paper do not bring much new for the scientific community. The English is hard to understand, due to unnecessary long sentences and grammatical errors. Finally, it contains a lot of figures from other papers: are the copyright issues well arranged? As significant changes are needed, I suggest now rejection.

Specific comments:

1. Introduction

-line 41: please define SMTZ here

-too long sentences are common, I suggest shorter ones for better understanding

-the aims and the novelty of this review are not well emphasised, please, try to correct this.

2. Methods

-line 79: what is etc here? please, clearly describe the used methods!

-please, indtroduce the used equipment, the location of the lab, type of equipment, calibrations, setup, etc.

3.Shape of pyrite

-Fig 2: is it sure, that all picture here is BSE image?

4. Texture of pyrite

-fig 4: is it sure, that all picture here is BSE image?

-a sub-chapter about the sunflower texture woud be appreciated here

5. Textural evolution

-line 215: I do not understand this 7th step here. You mentioned, greigite already turned to pyrite...

-lines 217-218: different or similar evolution? not clear

6. The texture of pyrite filling in organisms

-line 249: unpublished data of whom?

-fig 6: please, clearly state in figure caption, what kind of images are these? (BSE, SE ?)

-line 254: what kind of geochemical environments? how pyrite is linked to different environments? please, clarify!

7. Pyrite morphology in different geochemical zones in marine sediment

-line 265: why only the former two reflects to the redox conditions?

-lines 270-279: this paragraph is not clear, there are too many abbreviations and also the sentences are unclear. 

-subchapter 7.2. a similar summarising table, than the one in 7.1 would be useful

8. Alteration in later diagenetic process

-line 404: please, be aware of the use of different kinds of brackets

-lines 403-411: this paragraph contains maybe important new finding, but due to bad grammar it is really hard to understand. Please, correct it.

-fig 11C: is it sure, that it is a BSE image?

Author Response

  1. Introduction

-line 41: please define SMTZ here

-too long sentences are common, I suggest shorter ones for better understanding

-the aims and the novelty of this review are not well emphasised, please, try to correct this.

 Response: Thanks for the comments. The definition of SMTZ has been added as to the suggestion of the reviewer. And some long sentences have been already rewritten accordingly. This manuscript is a review about the morphology of authigenic pyrite, however, we did state the aims and novelty in line 74-76.

  1. Methods

-line 79: what is etc here? please, clearly describe the used methods!

-please, indtroduce the used equipment, the location of the lab, type of equipment, calibrations, setup, etc.

  Response: This section is a general introduction about the methods which have been used for morphological study in previous studies. To have the manuscript concise, the methods are only introduced generally without detail of experiment parameters, because such parameters might be different from lab to lab and from researcher to researcher.

3.Shape of pyrite

-Fig 2: is it sure, that all picture here is BSE image?

   Response: Thanks for pointing out this, the BSE and SE images are specified.

  1. Texture of pyrite

-fig 4: is it sure, that all picture here is BSE image?

 Response: Thanks for pointing out this, the BSE and SE images are specified.

-a sub-chapter about the sunflower texture woud be appreciated here

 Response: Thanks for your suggestion. A sub-chapter of sunflower has been added accordingly.

  1. Textural evolution

-line 215: I do not understand this 7th step here. You mentioned, greigite already turned to pyrite...

-lines 217-218: different or similar evolution? not clear

 Response:

Thanks for pointing out this. Indeed, the step 7 is not necessary, so it has been changed accordingly.

lines 217-218: Changed the expression to the similar textural evolution in several steps 

  1. The texture of pyrite filling in organisms

-line 249: unpublished data of whom?

 Response:Thanks for pointing out this. It has been corrected.

-fig 6: please, clearly state in figure caption, what kind of images are these? (BSE, SE ?)

-line 254: what kind of geochemical environments? how pyrite is linked to different environments? please, clarify!

 Response: Thanks for the comments about the geochemical environments in an organism chamber. The geochemical environments in organism chamber have been stated in the text.

The relation between pyrite morphology and different environment is discussed in detail in Section 7.

  1. Pyrite morphology in different geochemical zones in marine sediment

-line 265: why only the former two reflects to the redox conditions?

 Response: Because the former two are formed in the environments of either closing to sea floor or early diagenesis, the size of framboid is closely related to its formation environment. Although the morphology of pyrite could also be changed in other geological processes in late diagenesis, only the pyrite formed in very early diagenesis without reformation has implications for paleo condition.

-lines 270-279: this paragraph is not clear, there are too many abbreviations and also the sentences are unclear. 

 Response: Thanks for the comments. Some of the abbreviations have been replaced with the full expressions.

-subchapter 7.2. a similar summarising table, than the one in 7.1 would be useful

 Response: Thanks for the comments. We think an image can show the size differences clearly, and the original data can be found in the references, so to keep the review concise a table might not be necessary.

  1. Alteration in later diagenetic process

-line 404: please, be aware of the use of different kinds of brackets

Response: Thanks for pointing out this. The brackets have been corrected accordingly.

-lines 403-411: this paragraph contains maybe important new finding, but due to bad grammar it is really hard to understand. Please, correct it.

Response: Thanks for the comments. The paragraphs mentioned have been rephrased.

-fig 11C: is it sure, that it is a BSE image?

Fig 11C: Thanks for pointing out this. It should be SE images.